# A Multi-Disciplinary Team Approach to Genomic Testing for Drug-Resistant Epilepsy Patients—The GENIE Study

**DOI:** 10.3390/jcm11144238

**Published:** 2022-07-21

**Authors:** Lata Vadlamudi, Carmen Maree Bennett, Melanie Tom, Ghusoon Abdulrasool, Kristian Brion, Ben Lundie, Hnin Aung, Chiyan Lau, Jonathan Rodgers, Kate Riney, Louisa Gordon

**Affiliations:** 1The University of Queensland, UQ Centre for Clinical Research, Herston, QLD 4029, Australia; c.bennett@uq.edu.au (C.M.B.); melanietom7@gmail.com (M.T.); 2Department of Neurology, Royal Brisbane and Women’s Hospital, Herston, QLD 4029, Australia; 3Pathology Queensland, Royal Brisbane and Women’s Hospital, Herston, QLD 4029, Australia; ghusoon.abdulrasool@health.qld.gov.au (G.A.); kristian.brion@health.qld.gov.au (K.B.); ben.lundie@health.qld.gov.au (B.L.); hnin.aung@health.qld.gov.au (H.A.); chiyan.lau@health.qld.gov.au (C.L.); 4Faculty of Medicine, The University of Queensland, Herston, QLD 4029, Australia; jonathan.rodgers@health.qld.gov.au (J.R.); kate.riney@health.qld.gov.au (K.R.); 5Genetic Health Queensland, Royal Brisbane and Women’s Hospital, Herston, QLD 4029, Australia; 6Neurosciences Unit, Queensland Children’s Hospital, South Brisbane, QLD 4101, Australia; 7QIMR Berghofer Medical Research, 300 Herston Road, Herston, QLD 4006, Australia; louisa.gordon@qimrberghofer.edu.au; 8School of Public Health, The University of Queensland, Herston, QLD 4029, Australia; 9School of Nursing, Queensland University of Technology, Kelvin Grove, QLD 4059, Australia

**Keywords:** genomics, epilepsy, drug-resistant, diagnosis

## Abstract

Background. The genomic era has led to enormous progress in clinical care and a multi-disciplinary team (MDT) approach is imperative for integration of genomics into epilepsy patient care. Methods. The MDT approach involved patient selection, genomic testing choice, variant discussions and return of results. Genomics analysis included cytogenomic testing and whole exome sequencing (WES). Neurologist surveys were undertaken at baseline and after genomic testing to determine if genomic diagnoses would alter their management, and if there was a change in confidence in genomic testing and neurologist perceptions of the MDT approach. Results. The total diagnostic yield from all genomic testing was 17% (11/66), with four diagnoses from cytogenomic analyses. All chromosomal microarray (CMA) diagnoses were in patients seen by adult neurologists. Diagnostic yield for WES was 11% (7/62). The most common gene with pathogenic variants was *DCX*, reported in three patients, of which two were mosaic. The genomic diagnosis impacted management in 82% (9/11). There was increased confidence with integrating genomics into clinical care (Pearson chi square = 83, *p* = 0.004) and qualitative comments were highly supportive of the MDT approach. Conclusions. We demonstrated diagnostic yield from genomic testing, and the impact on management in a cohort with drug-resistant epilepsy. The MDT approach increased confidence in genomic testing and neurologists valued the input from this approach. The utility of CMA was demonstrated in epilepsy patients seen by adult neurologists as was the importance of considering mosaicism for previously undiagnosed patients.

## 1. Introduction

It is estimated that around 50 million people worldwide are affected with epilepsy [1]. Epilepsy, which manifests with seizures, has an enormous impact on patient quality of life and productivity [2], as well as substantive health costs [3,4]. More than one third of these patients are resistant to current anti-seizure medications [5], leading to even greater costs to the patient and health system [4,6]. Epilepsy is a heterogeneous group of disorders with an established genomic contribution [7,8,9]. For more than two thirds of epilepsy patients, where the cause is not otherwise established, genomic factors are likely to play a role [10].

From a management perspective, the benefits of genomic testing include earlier diagnosis [11] and more targeted precision medicine-based approaches [12,13,14]. The greatest yield from genomic testing is associated with early onset of seizures, drug-resistant epilepsy and the presence of developmental co-morbidity [15]. There is an increasing breadth of information regarding what genetic test should be undertaken to achieve a good diagnostic yield [11] and the practicing clinician is not always able to maintain contemporaneous knowledge. The finding of multiple variants of uncertain significance often leaves clinicians struggling to interpret and discuss the results with their patients [16]. 

A multi-disciplinary team (MDT) approach represents a pathway for better patient outcomes and potentially more cost-effective healthcare [17,18]. An ideal team approach would be choice of patient by the clinician, to determine if genomic testing will be clinically meaningful in terms of diagnosis and more targeted treatment possibilities [11]; choice of genomic testing and how to discuss the results by a clinical geneticist; and understanding of testing limitations and interpretation of pathogenicity by genetic pathologists and clinical variant curation scientists. Pre-test genomic counselling and a detailed discussion of returned genomic results, to ensure adequate post-test counselling, are also integral aspects for genomic testing [17,19]. 

The aim of the Group Engaged Next Generation Sequencing In Epilepsy patients (GENIE) study was to report diagnostic yield, the impact of a genomic diagnosis on management, and to ascertain if the MDT approach would increase genomic confidence for neurologists and determine their perception of this approach. 

## 2. Materials and Methods

### 2.1. Setting

Human Research Ethics Committee approval (HREC/2019/QRBW/54086) was obtained for a multi-centre project involving both children and adults. The implementation project was undertaken at three hospitals in the state of Queensland, Australia; Royal Brisbane and Women’s Hospital (RBWH), Queensland Children’s Hospital (QCH) and Cairns Base Hospital (CBH) and referrals were made prospectively (including both existing and newly presenting patients) over a 12-month period from January 2020 until December 2020. At RBWH, patients could be seen by adult neurologists from 15 years of age.

### 2.2. Target Population

Inclusion criteria were defined as patients with drug-resistant epilepsy, categorised as patients that had failed to become seizure free with adequate trials of two different anti-seizure medications. Patients with other neurological co-morbidities such as intellectual impairment, autism or cerebral palsy were included. Patients with a structural brain abnormality causing epilepsy, which can have an underlying genetic aetiology, such as malformations of cortical development, were included. 

Exclusion criteria included the presence of a non-genetic aetiology (such as an acquired brain injury); a genetic cause for epilepsy had been previously identified by genomic testing; patients who required rapid genomic testing for acute clinical care decisions; drug-resistant epilepsy phenotypes that lack well-established monogenic causes, such as the idiopathic generalised epilepsies. Some patients with complex multi-system phenotypes were referred to genetic services for consideration of broader genomic testing options.

Eligible patients were identified by their treating neurologist who completed a patient checklist (included as Supplementary Material), which included thorough phenotypic information regarding epilepsy and other clinical features. Developmental delay was used specifically for the developing child and could include multiple domains, whilst cognitive impairment was used for older patients, more specifically for intellectual impairment. Informed consent and a detailed family history were obtained by a genetic counsellor, who was part of the research team. Where possible, trio samples were sought for whole exome sequencing (WES), although duo and singleton samples were accepted.

If the patient had previous genomic testing, a discussion of each case was undertaken at the MDT meeting for consideration of inclusion. The patient was deemed not eligible for recruitment if previous genomic testing was undertaken in the last 5 years or review of the phenotype identified exclusion criteria.

### 2.3. Multi-Disciplinary Team

Our regular MDT meeting consisted of adult and paediatric neurologists, clinical geneticists, genetic pathologists, genetic counsellors, clinical variant curation scientists and our research team. Figure 1 highlights the steps in the MDT approach. The meeting format involved presentation of the phenotype, family tree, sample collected and genotype results. Return of results was dependent on the outcomes of the genomic testing and the treating neurologist decided who would return the results (either the research team, genetic counsellor or treating neurologist). Some patients were referred to clinical genetic services for further follow up after discussion at the MDT meeting.

### 2.4. Genomic Testing

Patients were required to have cytogenomic testing (extended karyotype and chromosomal microarray) completed prior to progressing to whole exome sequencing (WES). In some cases, these had already been completed as part of routine clinical care, so when these met appropriate quality thresholds the MDT agreed on a case-by-case basis that they did not need to be repeated. This included explicitly extended metaphase analysis for karyotype and a resolution limit of at least 200 kb for chromosomal microarray. In some cases where blood sampling caused significant distress, and sufficient stored DNA was available for WES, the MDT reviewed the phenotype and allowed extended karyotype exceptions if the phenotype did not fit reported phenotypes for chromosomal structural rearrangements.

### 2.5. Cytogenomic Testing

#### 2.5.1. Extended Karyotyping

Peripheral blood specimens were collected in lithium heparin tubes and lymphocytes were cultured in commercially available culture media supplemented with Phytohaemagglutinin at 37 degrees for 72 h. Genial Genomics CellSprint automated suspension metaphase harvester was used to harvest synchronised cultured lymphocytes as per manufacturer’s instructions. Fifteen G-banded metaphases are routinely analysed per specimen for karyotyping. For the current cohort, analysis was extended to 60 metaphases to detect low-level mosaicism.

#### 2.5.2. Fluorescence In Situ Hybridization

For the case with ring chromosome 5, Fluorescence in situ hybridization (FISH) was performed using the metasystems and Vysis ToTelVysion probes mix on unbanded metaphases. 

#### 2.5.3. Chromosomal Microarray Analysis (CMA)

DNA was extracted from EDTA preserved peripheral blood using the QIAGEN QIASymphony DNA Midi Kit or Chemagen blood 4K DNA Extraction kit. Patient’s DNA samples were amplified, fragmented, and hybridized to the Illumina beadchip array (CytoSNP 850K Array), followed by single base extension, in order to determine the genotype and copy number status for each locus as per the Illumina Infinium HD assay protocol. Results were analysed with BlueFuse Multi version 4.4, using genome reference sequence GRCh37/hg19. The effective resolution was 200 kb. The copy number variants that meet the internal laboratory reporting criteria were classified using the 2020 ClinGen/ACMG standards [20] for the interpretation and reporting of constitutional copy number variants.

If a Variant of Uncertain Significance (VUS) was identified from the CMA, the clinical significance was determined by assessing phenotypes of reported cases with similar copy number variants, the gene content, the correlation between those genes and the phenotype, and whether evidence existed to suggest that duplication or deletion of an encompassed gene was a pathogenic mechanism. In cases where the variant was plausibly related to the presentation, further evidence was sought through segregation studies.

### 2.6. Whole Exome Sequencing (WES)

#### 2.6.1. Sequencing and Bioinformatics Pipeline

DNA was extracted from EDTA-preserved whole blood using the QIAGEN QIAsymphony Midi Kit. Next generation sequencing and primary analysis (BCL-FASTQ) was performed either by the Australian Genome Research Facility or the Kinghorn Centre for Clinical Genomics Sequencing Laboratory using the Agilent Technologies SureSelect Clinical Research Exome V2 (CREv2); 67.3 Mb, sequenced on an Illumina NovaSeq 6000 or HiSeq X Sequencing System. The Bioinformatics Pipeline secondary analysis (FASTQ-BAM/VCF) was performed using DRAGEN Germline Pipeline software v3.2.8 and tertiary analysis (annotation and variant classification) completed using VarSeq (Golden Helix) software (v2.2.1). QC metrics were interrogated using Euformatics OmnomicsQ Software (v1.0.32.0) and in-house Sonar software. This analysis included coding sequences and at least ~10 bp of flanking intronic sequence. 

#### 2.6.2. Genes Analysed

Analysis was restricted to gene panels selected from Genomics England PanelApp (https://panelapp.genomicsengland.co.uk/, accessed on 11 February 2021). Only genes with strong clinical evidence for disease association (“green” genes) were used for analysis. The standard panels used were genetic epilepsy syndrome (v2.38) and malformations of cortical development (v2.5) and a total of 458 genes were analysed. Due to specific phenotypic features, two patients had additional panels, anophthalmia or microphthalmia panel in one and holoprosencephaly and septo-optic dysplasia panel for the other.

#### 2.6.3. Variant Filtration Chain

The filtering process included the gene of interest (GOI) filter using the panels described above to retain variants in these 458 genes. A variant impact filter was then applied to retain high and medium impact variants. Low impact variants (intronic and synonymous) where in silico splice tools within VarSeq (GeneSplicer, MaxEntScan, NNSplice and PMW) predicted disruption of a splice site (minimum 2 of 4) or the creation of a novel splice site (minimum 3 of 4) were also retained. Variants in the genome aggregation database (gnomAD) with a minor allele frequency (MAF) > 0.01 were excluded. For trio analyses, inheritance patterns were used to prioritise compound heterozygous or homozygous variants (autosomal recessive genes), de novo variants and heterozygous inherited variants in autosomal dominant genes where the gnomAD MAF was below 0.001. Variants with a read depth below 10× or alternate allele frequency below 0.15 were excluded from analysis. For singleton and duo samples, variant prioritization meetings were undertaken to further filter based on phenotype. Laboratory-based variant review meetings were undertaken prior to discussion at the MDT meeting.

#### 2.6.4. Variant Prioritization Meetings

For singleton and duo analyses, to reduce the burden of variant curation, variant prioritization meetings were conducted. These meetings included a neurologist, clinical geneticist, and senior variant curator. Low impact variants (based on in silico prediction and disease mechanism) and non-ClinVar listed pathogenic variants in genes associated with disorders that were not consistent with the patient’s clinical presentation were discarded.

#### 2.6.5. Variant Classification

Variants were described using Human Genome Variation Society (HGVS), v20.05 nomenclature. Variants were classified according to ACMG Guidelines [21], ACGS Guidelines [22] and SVI recommendations (https://www.clinicalgenome.org/working-groups/sequence-variant-interpretation/, accessed from 21 January 2021) using evidence available at the time of reporting. Single heterozygous variants detected in genes associated with autosomal recessive disorders were not routinely reported unless classified as likely pathogenic or pathogenic and considered relevant to the clinical indication. Any reportable likely pathogenic (class 4) or pathogenic (class 5) variants were confirmed by Sanger sequencing prior to reporting. Relatedness assessments using Kinship Coefficient were performed for data derived from family duo/trio samples.

#### 2.6.6. Laboratory Based Variant Review Meeting

The variant classification was then discussed in a laboratory-based variant review meeting, where the goal was to obtain consensus for criteria used and harmonise analyses. These meetings included variant curators and genetic pathologists.

#### 2.6.7. Multi-Disciplinary Meeting (MDT)

All results were then presented to the MDT meetings for discussion of findings prior to reporting. Discussion with treating neurologists was also encouraged where possible, to assist with classification, reporting and return of results.

### 2.7. Neurologist Survey

To assess the impact from the genomic testing, confidence in genomic testing and neurologist perception of the MDT approach, neurologists completed a survey at recruitment and after receiving the genomic results (see Appendix A).

## 3. Results

### 3.1. Total Patient Numbers (104)

Figure 2 shows the breakdown of patient referrals and number recruited. Of the 104 referrals, 24 declined (too busy, lost to follow-up, not interested) and five were deemed non-eligible, leaving 75 patients recruited.

### 3.2. Patients Recruited (75) 

The mean age at the time of recruitment into the study was 26 years (age range 2–71 years). The mean age of seizure onset was 11.6 years (range 0–61). There were 63% females (*n* = 47) and 37% males (*n* = 28) recruited. A total of 63% of patients were recruited from RBWH (*n* = 47); 28% from QCH (*n* = 21); and 9% from CBH (*n* = 7). 

Of these 75 patients, 56% (*n* = 42) had other neurodevelopmental abnormalities in addition to their epilepsy, which included cognitive impairment 39% (*n* = 29); developmental delay 29% (*n* = 22); speech impairment 16% (*n* = 12), autism spectrum disorder 12% (*n* = 9), attention deficit hyperactivity disorder 3% (*n* = 2); behavioural issues 7% (n = 5); and regression 15% (*n* = 11). Malformations of cortical development were identified on neuroimaging in 28% (*n* = 21) of patients.

After recruitment, three patients declined; three patients were excluded after MDT discussions as an acquired cause could not be excluded; and three patients were referred to the clinical genetic services due to complexity of their co-morbidities. This left 66 patients proceeding to genomic testing.

### 3.3. Diagnostic Yield from Genomic Testing (66)

Our total diagnostic yield from all genomic testing was 17% (11/66). Epilepsy classifications were undertaken by LV. Based on the 2017 International League Against Epilepsy classification (ILAE) epilepsy classification [23], the cohort had 59% (*n* = 39) focal epilepsy; 27% (*n* = 18) generalised epilepsy; and 14% (*n* = 9) with combined generalised and focal epilepsy. 

Of these 66 patients, 55% (*n* = 36) had other neurodevelopmental abnormalities in addition to their epilepsy, which included cognitive impairment 36% (*n* = 24); developmental delay 29% (*n* = 19); speech impairment 15% (*n* = 10), autism spectrum disorder 12% (*n* = 8), attention deficit hyperactivity disorder 3% (*n* = 2); behavioural issues 6% (*n* = 4); and regression 12% (*n* = 8).

#### 3.3.1. Cytogenomic Testing

CMA was performed on all 66 cases and extended karyotype was able to be obtained for 63 cases. Genomic diagnosis in relation to epilepsy phenotype was identified in three patients by CMA and one patient by extended karyotype (and FISH). CMA and extended karyotype results with a diagnosis are shown in Table 1 and include two recurrent copy number variants (22q11.21 duplication and 15q13.3 deletion), Xp21.3-p21.1 deletion, and a ring chromosome 5 syndrome. In the ring chromosome 5 case, the karyotype analysis showed a ring chromosome 5 and a small supernumerary marker chromosome (+mar) in all 30 metaphases examined. 

#### 3.3.2. Whole Exome Sequencing Testing (62)

In total, 62 patients progressed to whole exome sequencing (WES) and trio samples were able to be obtained in 54% (*n* = 34); duo samples in 23% (*n* = 14) and singleton samples in 23% (*n* = 14). A total of 53% (33/62) had associated neurodevelopmental co-morbidities and 34% (21/62) had neuroimaging supporting a malformation of cortical development. Neurodevelopmental co-morbidities were associated with 29% (8/36) in focal epilepsy; 94% (16/17) in generalised epilepsy; and 100% (9/9) in combined focal and generalised epilepsies. Some form of non-diagnostic genomic testing (single gene testing or limited gene panel) had been previously undertaken in 13% (*n* = 8) of patients.

Reportable variants were identified in 11 patients with seven of these being diagnostic (class 4/5) and five being non-diagnostic or variants of uncertain significance (class 3). Our diagnostic yield for these 62 patients was 11% (7/66) and when analysed by epilepsy type was 6% (2/36) for the focal epilepsy; 12% (2/17) for generalised epilepsy; and 33% (3/9) for the combined focal and generalised epilepsies. Reportable variants from WES are shown in Table 2.

There were 19 patients who had a total of 29 VUS findings discussed at MDT meetings. Of these 29 VUS, a total of five were deemed reportable based on the likelihood of the variant being causative and clinically actionable if reported. These five VUS were identified in the following genes—*CHRNA2, TSC2, NPRL3* in two cases and *POLG*.

### 3.4. Variant Prioritization Meetings

Variant prioritization meetings were performed for the duo (*n* = 14) and singletons (*n* = 14). Variants were prioritized for curation if the gene in which they were identified could explain the presentation, was not seen in control populations, and the type of variant was consistent with the known molecular mechanism of disease. For the whole group (*n* = 28), 69% of variants (260/379) were discarded after this discussion and 31% went on to variant curation.

For duos, 64% (108/168) were discarded, and for singletons 72% (152/211) were discarded with 36% and 28%, respectively, going onto variant curation. The average number of variants for curation was 4.75 (trios), 5 (duos) and 4.54 (singletons).

### 3.5. Impact to Management of a Genomic Diagnosis (11)

Of the 66 patients undergoing genomic testing, 62 neurologist surveys with both baseline and after genomic results were obtained (12 neurologists).

For the cohort who had a genomic abnormality identified, 82% (9/11) of the neurologists felt that the genomic diagnosis impacted their management. Neurologists stated they were planning to change anti-seizure medications (ASM) for four patients. Comments for these four patients included further review of the literature (mosaic *DCX*); an increase in ASM due to ongoing seizures (Xp21 deletion); decision to use a genetic ASM such as valproate (*YWHAG*); and use a genetic ASM such as valproate and cease vigabatrin (Ring chromosome 5). There was one patient where the neurologist was reconsidering valproate after the genomic results (mosaic *DCX,* and heterozygous POLG VUS).

Other management decisions were changed in two patients, resective surgical workup would cease for one and deep brain stimulation to be considered (mosaic *DCX)* and the other was deemed not for surgical consideration (*NPRL3*). There were four patients where fewer investigations would be performed in the future and comments included less likelihood to repeat investigations in the case of refractory seizures, such as MR brain (15q13.3 microdeletion); no SPECT/PET/further imaging, no genome/exome sequencing (Ring chromosome 5); no further investigations for surgical assessment (mosaic *DCX)*; and not for VEM/PET (NPRL3).

For the patients with the other four VUS, functional genomics analyses have been undertaken in two cases (*CHRNA2* and *NPRL3*) to ascertain if the genomic diagnoses can be re-reviewed in the future.

### 3.6. Multi-Disciplinary Team Approach

Table 3 demonstrates changes in moderate confidence ability for different aspects of genomic testing from baseline to after genomic testing. Neurologists were also asked to rank how comfortable they felt with integrating genomics into clinical care on a rating scale from 1 (least confident) to 7 (most comfortable), at baseline and after genomic testing. The outcome demonstrated a clear shift in the 7-point rating score from those scoring 5–7 at baseline for 66% (41/62) to 94% (58/62) after genomic testing (Pearson chi square = 83, *p* = 0.004).

A total of 53% of patients had their treating neurologist (33/62) attend the MDT and qualitative comments of what they found useful about the MDT were included in Table 3.

## 4. Discussion

### 4.1. Diagnostic Yield of Genomic Testing

Our total diagnostic yield was 17% including both cytogenomic testing and WES. We demonstrated abnormal cytogenomic testing (CMA and extended karyotype) in four patients, emphasizing the importance of careful patient selection. Although the cost and availability play an important role in selecting the type of genomic testing, prioritising cytogenomic over WES testing in a clinical diagnostic setting should be mainly driven by the clinical assessment, specificity of the phenotype and the likely diagnosis.

### 4.2. Cytogenomic Testing

The pathogenic Xp21 deletion detected in case 1 encompasses four haploinsufficient (ClinGen HI score:3) genes and is associated with Xp21 deletion syndrome (OMIM 300679). Most heterozygous female carriers of Xp21 deletion are asymptomatic, and clinical manifestations (cognitive impairment, developmental delay, epilepsy, and mild features of muscular dystrophy) as seen in this patient are due to X chromosome inactivation [24].

The recurrent pathogenic 22q11.2 duplication (proximal, A–D) detected in case 2 is associated with reduced penetrance, and variable expressivity and inheritance from apparently unaffected parents have been reported in the literature [25]. In view of this, inheritance from a clinically unaffected mother does not exclude pathogenicity of this CNV. The 22q11.2 duplication is likely contributing to this patient’s symptoms.

The 15q13.3 recurrent pathogenic deletion detected in case 3 was a smaller deletion than typically observed (D-CHRNA7 to BP5) nested within the 15q13.3 recurrent (BP4-BP5) microdeletion region but encompasses the CHRNA7 gene and has a Clingen Haploinsufficiency (HI) score of 3.

The ring 5 chromosome detected in case 4 showed evidence of mitotic instability and dynamic mosaicism associated with ring syndrome which is characterized by growth retardation (short stature, microcephaly), cognitive impairment, and mild anomalies. The ring syndrome phenotype is highly variable due to the extent of aneuploidy and level of mosaicism in various tissues, however, it was consistent with this patient’s phenotype [26,27].

These findings reflect the importance of consideration of these tests by clinicians treating adult patients, who are often not as familiar with these types of genomics testing in routine practice. In our cohort, all CMA diagnoses were made in patients seen by adult neurologists with the majority having associated cognitive impairment. This point is further highlighted by Borlot and colleagues, who demonstrated that 16.1% of patients of their adult cohort with unexplained childhood epilepsy and intellectual disability had pathogenic or likely pathogenic CMA results [28].

### 4.3. Whole Exome Sequencing

Our diagnostic yield from WES was 11%. A large meta-analysis investigation demonstrated a diagnostic yield of 9.3% with epilepsy alone and 27.9% for epilepsy with intellectual disability [29]. The diagnostic yield from Johannesen and colleagues [30] was 23% in a cohort aged from 18 to 80 years. Their cohort had a higher percentage with co-morbidities (91%) compared with our cohort (53%), which may account for their higher yields. Similarly, Borlot and colleagues demonstrated a diagnostic yield of 22% in adult epilepsy patients with intellectual disability [31]. Whilst our patient numbers were small, we observed increased neurodevelopmental co-morbidities and diagnostic yield in the combined generalised and focal epilepsies, in line with the meta-analysis [29].

In the paediatric patients (cases 5, 6 and 7), our yield included known gene variants associated with developmental and epileptic encephalopathy. Pathogenic variants in the *DCX* gene have typically been associated with X-linked subcortical laminar or band heterotopia in females and more severe X-linked lissencephaly in males. However, in our cohort, all three patients (cases 8, 9 and 11) including one male patient with pathogenic variants in the *DCX* gene presented with clinical features and MRI findings consistent with subcortical laminal heterotopia. The milder phenotype in the male patient is explained by the presence of a mosaic pathogenic variant in the *DCX* gene.

Of particular interest in our cohort, two unrelated cases (cases 9 and 11) had an identical de novo pathogenic *DCX* variant (c.556C>T, p.(Arg186Cys)) and in both cases the variant was mosaic (variant allele frequencies of 13% and 24%). This variant is present in ClinVar and was one of the first *DCX* variants described in literature [32]. One potential mechanism for de novo occurrence could be because of the location of the variant in a cytosine-guanine (CpG) dinucleotide, causing a mutation hotspot due to spontaneous deamination of 5-methylcytosine, producing guanine to thymine mismatches [33].

In case 11, previous Sanger sequencing of the *DCX* gene failed to identify the variant, due to the low variant allele frequency (13%). With our standard whole exome sequencing process, initially, the *DCX* variant was also filtered out, but because of the ability to provide clinical input to the clinical variant curation scientist, manual and visual inspection of the data using integrative genomic viewer (IGV) was undertaken, which identified the variant. This case highlights the success of our MDT approach on the genomic outcome.

A case report similarly has demonstrated the utility of exome sequencing to detect mosaic variants at an allele frequency of 18% in another malformation of cortical development gene (*LIS1*) [34]. The case highlights the importance of considering malformation of cortical development genes, and in particular mosaicism, in patients with drug-resistant epilepsy. Mosaic pathogenic variants in the *DCX* gene are associated with milder presentation and mainly seen in individuals with subcortical band heterotopia [35].

The other interesting gene identified was *NPRL3* (case 10), which involves the mTOR pathway and an increasingly important gene identified in the drug-resistant focal epilepsies. A likely pathogenic variant (c.1300delG, p.(Val434Serfs*23)) in the *NPRL3* gene was identified in one patient in our cohort with focal epilepsy. There was no family history of epilepsy. Pathogenic loss of function variants in the *NPRL3* gene have been reported in patients and families with focal epilepsy [36,37,38]. The phenotypic spectrum associated with the *NPRL3* gene ranges from severe drug-resistant epilepsy associated with developmental impairment to mild epilepsy and infrequent seizures [39]. Additionally, there is significant incomplete penetrance, with many unaffected pathogenic variant carriers reported within a family. The value of identification of this variant is to search for a subtle cortical malformation that could be surgically amenable; risk prediction for relatives; reproductive counselling; and the potential for targeted treatment approaches with mTOR inhibition [40].

### 4.4. Impact to Management of a Genomic Diagnosis

Contributing to our high proportion reporting impact to management (82%) is likely the diagnosis to patients after years of a diagnostic odyssey, enabling closure for the patients [41,42]. A cross-sectional study of 798 drug-resistant adult epilepsy patients demonstrated a diagnostic yield of 13.5%, and 57.4% had clinically actionable genetic findings [43]. The importance of genetic findings in therapeutic decision making was highlighted by Johannesen and colleagues where 17% (11/46) of patients had gene-specific treatment changes [30].

### 4.5. Multi-Disciplinary Team Approach

The reduction in the number of variants required for curation by more than two-thirds after the variant prioritization meetings indicates how input from clinicians with different expertise (neurologist, geneticist) can potentially improve service provision.

The increase in confidence and the qualitative comments from neurologists support the MDT approach to manage the growing complexity of genomic testing [44]. During the study we observed an increase in engagement and neurologist referrals, demonstrated by greater attendance at MDT meetings, not only for patient discussions, but also for their genomic education.

At the MDT results were discussed with the treating neurologist and a comprehensive plan for return of results was made. Discussions included further testing considerations if results were negative or non-informative (VUS), or if referral to clinical genetic services or reproductive risk counselling were required. To ensure optimal return of results for the patient, the meeting provided a forum for the neurologist to understand the findings and discuss implications with the MDT, in advance of the patient’s appointment [45].

### 4.6. Limitations

The small sample size and the selection bias, with inclusion of patients referred from a cohort of 12 neurologists from three sites, are limitations. The cohort had a wide breadth of ages, with a clear bias to more adult patients. The numbers of patients in the sub-groups such as focal, generalised and combined generalised and focal epilepsies were small for analyses, but the trends were in keeping with larger studies [29]. Our current whole exome sequencing was limited in the identification of duplication and deletions, which may contribute to a lower yield. A higher yield was demonstrated in a cohort with whole genome sequencing after non-diagnostic exome sequencing [46]. We were also limited in the number of trios obtained as many of our patients were older. Ideally, longer follow-up after genetic testing would be required to assess the economic benefits of the genetic testing in terms of reduction in investigations, changes to treatment and reduced hospital visits.

## 5. Conclusions


The diagnostic yield from all genomic testing was 17% in our cohortManagement was impacted in nine out of 11 patients with a genomic diagnosisThe MDT approach increased neurologist confidence in genomic testingNeurologists valued the input from the MDT approachRole of chromosomal microarray in patients with epilepsyMosaicism is important to consider in previously undiagnosed patients


The genomic era provides an exciting opportunity to advance care for the epilepsy patient, but this enormous breadth of information needs to be harnessed and an MDT approach maximizes benefit to patient care.

## Figures and Tables

**Figure 1 jcm-11-04238-f001:**
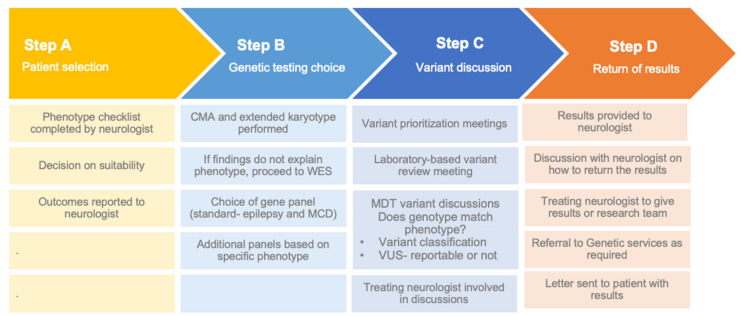
Steps in the multi-disciplinary approach to genomic testing. CMA—chromosomal microarray; WES—whole exome sequencing; MCD—malformations of cortical development; MDT—multi-disciplinary team.

**Figure 2 jcm-11-04238-f002:**
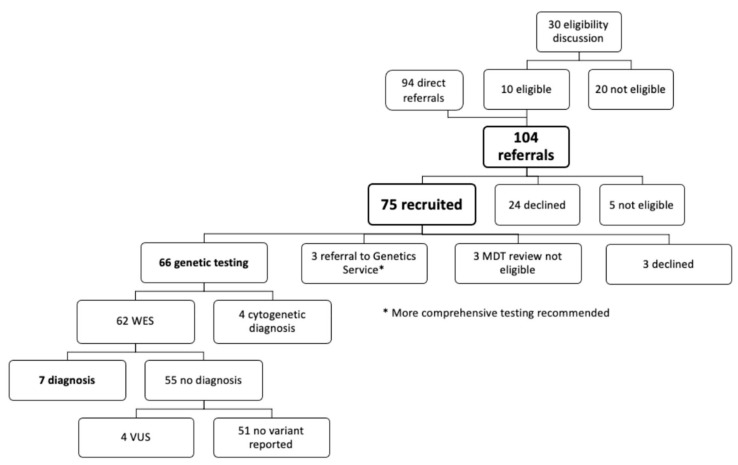
Patient referrals, number recruited, and genomic results. MDT-multi-disciplinary team; WES—whole exome sequencing; VUS-variant of uncertain significance.

**Table 1 jcm-11-04238-t001:** Description of the four cases where the genomic diagnosis was obtained using cytogenomic testing (CMA or extended karyotype).

Case	Sex	Seizure Onset (Years)	Age at Study (Years)	Epilepsy	Co-Morbidities	MR Brain	Abnormalities Detected	Size (Mb)	Assay	Inheritance	Diagnosis
1	F	20	32	Focal	Dysmorphic, DD, muscle weakness, CI	Slight prominence of ventricles	arr[GRCh37] Xp21.3p21.1(28725114_33610311)x1 dn	4.9	CMA	Assumed de novo **	Xp21 deletion syndrome (MIM 300679)
2	F	10	17	Focal	Nil	NAD	arr[GRCh37] 22q11.21(18844632_21463730)x3 mat	2.6	CMA	Maternal	22q11.2 microduplication syndrome (MIM 608363)
3	F	13	16	Generalised	Mild CI	NAD	arr[GRCh37] 15q13.3(32019919_32514341)x1	0.5	CMA	N/A	15q13.3 microdeletion syndrome (MIM 612001)
4	F	9	11	Generalised	Dysmorphic, CI, microcephaly	NAD	47,XX,r(5)(q15.?3q3?2),+mar[27]/47,XX,dup r(5),+mar[3] *	-	EK	N/A	Ring chromosome 5 syndrome

All variants were described using Genome Reference Consortium Human Build 37 (GRCh37). CMA—chromosomal microarray analysis; EK—extended karyotype; DD—developmental delay; CI—cognitive impairment; NAD—no abnormality detected; N/A—not available. * FISH using D5S1518E (5p15.2)/EGR1(5q31)/RPS14(5q32) probe set showed complex profile indicative of dynamic mosaicism and using C84c11/T3(5p)/D5S2907(5q) subtelomeric probe set showed the small supernumerary marker chromosome derived from chromosome 5. ** Mother was tested, and the copy number variation was non-maternal in origin. De novo inheritance is assumed as the father was unaffected and not tested.

**Table 2 jcm-11-04238-t002:** Description of the seven cases where the genomic diagnosis was obtained using whole exome sequencing.

Case	Sex	Age of Onset	Age at Study (Years)	Epilepsy	Co-Morbidities	MR Brain	Sample	Gene Name	Variant Description/ClinVar ID	ACMG Classification/Criteria	Parental Origin/Inheritance	Diagnosis
5	F	6 months	6	Generalised	Dysmorphic, DD, CP	lissencephaly	Duo	*PAFAH1B1*	NM_000430.3:c.657G>A, p.(Trp219Ter), Chr17(GRCh37):g.2576037G>A (ID: 159533)	Pathogenic PVS1, PM2, PP4	Unknown (non-maternal) AD	Lissencephaly 1
6	F	14 months	5	Generalised	Dysmorphic, DD, CI	NAD	Trio	*YWHAG*	NM_012479.4:c.394C>T, p.(Arg132Cys), Chr7(GRCh37):g.75959244G>A (ID: 438804)	Pathogenic PS2, PS4_Moderate, PM1_Supporting, PM2, PP3	De novo AD	Developmental and epileptic encephalopathy 56
7	M	1 day	14	Generalised and Focal	Global DD, visual impairment, CP	Bilateral peri-sylvian poly-microgyria	Trio	*SCN2A*	NM_001040142.2:c.1129T>A, p.(Leu377Ile), Chr2(GRCh37):g.166170224T>A (ID: 1025193)	Likely Pathogenic PS2_Moderate, PM1, PM2, PP3, PP2	De novo AD	Developmental and epileptic encephalopathy 11
8	F	14 years	29	Generalised	CI	Band heterotopia	Trio	*DCX*	NM_001195553.2:c.580G>C, p.(Ala194Pro), ChrX(GRCh37):g.110644343C>G (Novel)	Pathogenic PS2, PM2, PM1, PP3, PP4	De novo XL	X-linked subcortical laminal heterotopia
9	M	12 years	30	Generalised	DD, mood disturbance	Band heterotopia	Duo	*DCX*	NM_001195553.2:c.556C>T, p.(Arg186Cys), ChrX(GRCh37):g.110644367G>A (ID: 158476)	Pathogenic PS2_Very strong, PS4_Moder-ate, PM2, PM1, PP3	De novo Mosaic VAF 24% XL	X-linked subcortical laminal heterotopia
10	F	13 years	44	Focal	Nil	NAD	Duo	*NPRL3*	NM_001077350.3:c.1300delG,p.(Val434Serfs*23), Chr16(GRCh37):g.139764delC (Novel)	Likely Pathogenic PVS1, PM2	Unknown (non-maternal) AD	AD familial focal epilepsy with variable foci 3
11	F	3 years	20	Generalised and Focal	CI	Band heterotopia	Trio	*DCX*	NM_001195553.2:c.556C>T, p.(Arg186Cys) ChrX(GRC37):g.110644367G>A (ID: 158476)	Pathogenic PS2_Very strong, PS4_Moderate, PM2, PM1, PP4, PP3	De novo Mosaic VAF 13% XL	X-linked subcortical laminal heterotopia

All variants were described using Genome Reference Consortium Human Build 37 (GRCh37); DD—developmental delay; CI—cognitive impairment; CP—cerebral palsy; AD—autosomal dominant, XL—X linked, VAF—variant allele frequency; ACMG—American College of Medical Genomics guidelines, NAD—no abnormality detected.

**Table 3 jcm-11-04238-t003:** Neurologist surveys- Moderate confidence ability for different aspects of genomic testing from baseline to after genomic testing as well as qualitative comments regarding the MDT approach.

Aspect of Genomic Testing	Moderately Confident	
	Baseline	After Genomic Test
Ability to interpret genomic results	44% (n = 27)	61% (n = 38)
Ability to explain genomic concepts	47% (n = 29)	68% (n = 42)
Ability to make treatment recommendations	37% (n = 23)	60% (n = 37)
Ability to provide genetic counselling	31% (n = 19)	42% (n = 26)
Ability to arrange the most appropriate genetic testing	34% (n = 21)	55% (n = 34)

**Qualitative comments by Neurologists regarding the multi-disciplinary meeting**		
To help me interpret genomic testing	Clarify no genomic cause	
Very helpful in understanding the process and uncertainties of testing	Helped me understand the complexity and meaning of variants	
Very informative	Group discussion	
Education	Helped me understand the complexities of genomic information	
Discussion of karyotype with lab input very useful	MDT interactions are informative and fulfilling	
Good to know	informative	
Education	Helped with understanding of the mutation and explaining to parents	
Discussion of the phenotype-genotype relationship invaluable	Understanding implications of findings	
Broad discussion with people of different expertise	Only went to pre-test interview—was very helpful	
The discussion afterwards was very helpful	Multi-disciplinary approach	

## Data Availability

Data are available on request.

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
