# Peer review of "A Multi-Disciplinary Team Approach to Genomic Testing for Drug-Resistant Epilepsy Patients—The GENIE Study"

_jcm, 2022, doi:10.3390/jcm11144238_

Round 1
Reviewer 1 Report
The authors report a team approach in which multi-disciplinary experts discuss clinical and genomic data in order to provide accurate diagnoses, in pediatric and adult patients, and to better manage patients with drug-resistant epilepsy. The manuscript is straightforward, nonetheless, could be improved if the authors consider the following points.
1. The first section of the abstract ("background") is written more like "aims/objectives/purpose" than a "background/introduction".
2.The authors declare to perform regular MDT meetings with adults and pediatric neurologists.
In the discussion, the authors stated that all the CNVs were detected in adult patients (line 339) but case 2 (17 y.o.), case 3 (16 y.o.) and case 4 (11 y.o.) could be considered pediatric patients. In section 4.2.1 Whole exome sequencing the authors reported that gene variants associated with DEE were detected in pediatric patients, in specific case SCN2A is a 14-year-old male. Please define better the age range used in this study to classify an individual as a pediatric or as an adult subject.
3. The diagnostic yield’s percentage of WES is written in the abstract and in the discussion, but it is not written in the corresponding result section.
4. It is interesting to know the comorbidities percentage in the 75 recruited individuals, but perhaps, it would also be interesting to highlight the distribution of the comorbidities among the 66 individuals for whom CMA and WES were performed.
5. In the 4.2-Cytogenomic testing paragraph, the authors discuss the two CMA results , the 22q11.2 duplication and the 15q13.3 deletion, but the Xq21 deletion and the ring chromosome 15 are not mentioned or commented on. In particular, it seems that case 4 has a phenotype slightly different from the cases reported in literature.
6. In the section 4.2.1-WES, the authors stated that the mosaic DCX variant (p.Arg186Cys) detected in the male proband could be responsible for his milder phenotype. The proband carries also a pathogenic variant in the POLG gene. Is it possible that the POLG alteration modulates or contributes to the patient's phenotype?
7. The authors reported to neurologist 4 VUS variants as they were considered likely clinically actionable, but there aren't information. Did these clinically actionable variants impact patient management or have changed pharmacological therapy?
8. Figure 1. The steps used in the MDT approach are clear, but please define the acronym MCD and the meaning of the asterisks used in step C.
9. Figure 2. The authors well described the criteria for patient inclusion/exclusion and the eligible patients. The 104 referrals are the total patients eligible to be studied after filling of the checklist, but it is not so clear how the “30 eligibility discussion” patients (10 eligible + 20 not eligible) were recruited, please clarify this step.
10. Table 2. Some suggestions:
A) The table title should be similar to the title of table 1.
B) Provide a progressive number for the listed variants as it is done for table 1 (example case 5 for patient with PAFAH1B variant; case 6: YWHAG variant;….).
C) Provide a rs code or specify if the variant detected is a novel variant. For example the DCX variant is recurrent and probably it has a dbSNP/ClinVar code.
D) Check the SCN2A protein nomenclature.
E) I would recommend specifying the ACMG criteria used to classify the variants.
F) lines 272-273: The legend under the table report “Reportable variants were identified in 11 patients with seven of these being diagnostic (class 4/5) and four being non-diagnostic (class 3).”, but table 2 reports the WES results of 7 LP/P cases and there are information about the 4 "class 3" variants.
11. lines 164, 217, 223, 261 and 344: It is not clear if these are section/subsection titles.
12. lines 217,223, 238, 261, 288: the sample number (n=...) should be removed.
13. line 230: the word "autism" should be replaced by "autism spectrum disorder"
14. line 281: specify the “significant number” used as cutoff for excluding a variant in the prioritization process.
15. lines 392,400 and 416: correct number headings
Author Response
The authors report a team approach in which multi-disciplinary experts discuss clinical and genomic data in order to provide accurate diagnoses, in pediatric and adult patients, and to better manage patients with drug-resistant epilepsy. The manuscript is straightforward, nonetheless, could be improved if the authors consider the following points.
- The first section of the abstract ("background") is written more like "aims/objectives/purpose" than a "background/introduction".
Thank you for this feedback and it has been changed to reflect background
2.The authors declare to perform regular MDT meetings with adults and pediatric neurologists.
In the discussion, the authors stated that all the CNVs were detected in adult patients (line 339) but case 2 (17 y.o.), case 3 (16 y.o.) and case 4 (11 y.o.) could be considered pediatric patients.
In section 4.2.1 Whole exome sequencing the authors reported that gene variants associated with DEE were detected in pediatric patients, in specific case SCN2A is a 14-year-old male. Please define better the age range used in this study to classify an individual as a pediatric or as an adult subject.
Our definition was based on who the patient was under the care of. At our hospital , patients older than 15 years are seen at the adult clinic. This has been added in line 84. In the table-3 out of the 4 patients were seen by adult neurologists (all the abnormal CMA results), the other patient with the abnormal extended karyotype was seen by a paediatric neurologist (11 yo). The reflections were more that adult neurologists were performing these tests rather than paediatrics neurologists, who are often more familiar with these tests. We have added a line to better clarify this. The 14 year old boy has onset of seizures in infancy but had WES at age 14 years and was under the care of a paediatric neurologist.
The abstract has been accordingly changed to reflect this,
- The diagnostic yield’s percentage of WES is written in the abstract and in the discussion, but it is not written in the corresponding result section.
This had been added line 278
- It is interesting to know the comorbidities percentage in the 75 recruited individuals, but perhaps, it would also be interesting to highlight the distribution of the comorbidities among the 66 individuals for whom CMA and WES were performed.
This has been added lines 282-285.
This has been added to the results
- In the 4.2-Cytogenomic testing paragraph, the authors discuss the two CMA results , the 22q11.2 duplication and the 15q13.3 deletion, but the Xq21 deletion and the ring chromosome 15 are not mentioned or commented on. In particular, it seems that case 4 has a phenotype slightly different from the cases reported in literature.
Information has been added to the discussion lines 451-460
- In the section 4.2.1-WES, the authors stated that the mosaic DCX variant (p.Arg186Cys)detected in the male proband could be responsible for his milder phenotype. The proband carries also a pathogenic variant in the POLG gene. Is it possible that the POLG alteration modulates or contributes to the patient's phenotype?
The patient’s DCX variant accounts for all of his features, and his phenotype is consistent with other males with mosaic pathogenic DCX variants in whom POLG variants were not identified. The phenotype associated with monogenic POLG variants would not account be a strong match for this patient’s features, and it is therefore more likely to represent an incidental finding.
The patient’s DCX variant accounts for all of his features, and his phenotype is consistent with other males with mosaic pathogenic DCX variants in whom POLG variants were not identified. The phenotype associated with monogenic POLG variants would not account be a strong match for this patient’s features, and it is therefore more likely to represent an incidental finding.
As the inclusion of the POLG variant makes the Table confusing, we have removed this from the table and included the variant in the list of VUSs
- The authors reported to neurologist 4 VUS variants as they were considered likely clinically actionable, but there aren't information. Did these clinically actionable variants impact patient management or have changed pharmacological therapy?
These variants have been added line 374-375. The further action undertaken is for further functional genomics analyses in 2 cases. A line on this is included in the impact to management at 405-407
- Figure 1. The steps used in the MDT approach are clear, but please define the acronym MCD and the meaning of the asterisks used in step C.
All acronyms have been added to the manuscript. Line 123-124
- Figure 2. The authors well described the criteria for patient inclusion/exclusion and the eligible patients. The 104 referrals are the total patients eligible to be studied after filling of the checklist, but it is not so clear how the “30 eligibility discussion” patients (10 eligible + 20 not eligible) were recruited, please clarify this step.
The patients were discussed at the MDT and this has been clarified in line 109-111
- Table 2. Some suggestions:
- A) The table title should be similar to the title of table 1.
This has been amended- line 322
- B) Provide a progressive number for the listed variants as it is done for table 1 (example case 5 for patient with PAFAH1B variant; case 6: YWHAG variant;….).
This has been added, thank you
- C) Provide a rs code or specify if the variant detected is a novel variant. For example the DCX variant is recurrent and probably it has a dbSNP/ClinVar code.
This has been added to the Table
- D) Check the SCN2A protein nomenclature.
SCNA2 Protein nomenclature corrected. A transcription error in the table only (377 not 277).
- E) I would recommend specifying the ACMG criteria used to classify the variants.
This has been added to the table
- F) lines 272-273: The legend under the table report “Reportable variants were identified in 11 patients with seven of these being diagnostic (class 4/5) and four being non-diagnostic (class 3).”, but table 2 reports the WES results of 7 LP/P cases and there are information about the 4 "class 3" variants.
We agree and have removed the line about the 4 class 3 variants and this has been discussed in the result section- lines 372- 375, the POLG has been added to this group and a total of 5 now
- lines 164, 217, 223, 261 and 344: It is not clear if these are section/subsection titles.
These have been amended, thank you
- lines 217,223, 238, 261, 288: the sample number (n=...) should be removed.
These have all been removed
- line 230: the word "autism" should be replaced by "autism spectrum disorder"
Thank you, this has been corrected line 260 (updated manuscript)
- line 281: specify the “significant number” used as cutoff for excluding a variant in the prioritization process.
Single value cut-offs (more conservative) for minor allele frequency of variants in gnomAD are built into the filter chain but are understandably on the conservative side so as not to miss anything. For the variant prioritization meetings, further variants could be excluded by the team (clinical gen., scientist, neurologist) based on expert knowledge and the information we had available and this was generally done on a per variant basis (gene-dependent), with no single figure available to be provided.
For example, a variant occurring in a gene associated with AD severe phenotype and infantile onset, but the variant is observed in multiple ‘normal’ adult individuals in gnomAD, then would be discarded. Similarly for a severe rare recessive condition, if said variant was observed in the homozygous state in gnomAD and at high frequency for heterozygotes it could be discarded.
Expert judgement by the team was used here, something that is difficult to replicate with a filter chain. These variants would very likely have BS1 and/or BS2 applied if curated in the lab and get to likely benign/benign, or very weak VUS at worst, so there is significant time saved if thrown out here.
We have removed the “in significant numbers” to avoid confusion
- lines 392,400 and 416: correct number headings
This has been corrected, thanks so much
Reviewer 2 Report
The authors around Vadlamudi et al. describe a well-designed approach for genetic testing of pharmacorefractory epilepsy patients, which is very clear and suitable for everyday practice with epilepsy patients. In doing so, an approach is described in a precise and well-structured manner that, although it is standard in many larger clinics anyway, is all the more practical and relevant for it. A particular strength is the clear and honest presentation of the results from real hospital life. On the other hand, this approach also brings the weakness that the figures are limited to a few Australian centers, only a few patients were included and the recruitment remains somewhat heterogeneous due to the approach, which the authors, however, honestly discuss under "Limitations". In addition, originality lags somewhat in terms of diagnostic workup with gene panels, as in many places whole exome or whole genome sequencing is already offered as standard in larger centers.
Further discussion of the results in terms of diagnostic yield in non-diagnostic exome sequencing or gene panels would be desirable (e.g., Palmer et al. 2021, Neurology, PMID: 33568551). In addition, the discussion would benefit from embedding the results in relation to larger systematic reviews and meta-analyses (e.g., Stefanski et al. 2021, Epilepsia, PMID: 33200402).
Author Response
The authors around Vadlamudi et al. describe a well-designed approach for genetic testing of pharmacorefractory epilepsy patients, which is very clear and suitable for everyday practice with epilepsy patients. In doing so, an approach is described in a precise and well-structured manner that, although it is standard in many larger clinics anyway, is all the more practical and relevant for it. A particular strength is the clear and honest presentation of the results from real hospital life. On the other hand, this approach also brings the weakness that the figures are limited to a few Australian centers, only a few patients were included and the recruitment remains somewhat heterogeneous due to the approach, which the authors, however, honestly discuss under "Limitations". In addition, originality lags somewhat in terms of diagnostic workup with gene panels, as in many places whole exome or whole genome sequencing is already offered as standard in larger centers.
Thank you for your comments and we agree with your thoughts
Further discussion of the results in terms of diagnostic yield in non-diagnostic exome sequencing or gene panels would be desirable (e.g., Palmer et al. 2021, Neurology, PMID: 33568551). In addition, the discussion would benefit from embedding the results in relation to larger systematic reviews and meta-analyses (e.g., Stefanski et al. 2021, Epilepsia, PMID: 33200402).
Thank you for these excellent article, these 2 publications have been included in the manuscript.